

# Different environmental gradients affect different measures of snake β-diversity in the Amazon rainforests

Rafael de Fraga[1,2,3], Miquéias Ferrão[1], Adam J. Stow[2],
William E. Magnusson[4] and Albertina P. Lima[4]

[1] Programa de Pós-Graduação em Ecologia, Instituto Nacional de Pesquisas da Amazônia, Manaus, Amazonas, Brazil
[2] Department of Biological Sciences, Macquarie University, Sydney, NSW, Australia
[3] Programa de Pós-graduação em Sociedade, Natureza e Desenvolvimento, Universidade Federal do Oeste do Pará, Santarém, Pará, Brazil
[4] Coordenação de Biodiversidade, Instituto Nacional de Pesquisas da Amazônia, Manaus, Amazonas, Brazil

Corresponding author
Rafael de Fraga,
r.defraga@gmail.com

## ABSTRACT

Mechanisms generating and maintaining biodiversity at regional scales may be evaluated by quantifying β-diversity along environmental gradients. Differences in assemblages result in biotic complementarities and redundancies among sites, which may be quantified through multi-dimensional approaches incorporating taxonomic β-diversity (TBD), functional β-diversity (FBD) and phylogenetic β-diversity (PBD). Here we test the hypothesis that snake TBD, FBD and PBD are influenced by environmental gradients, independently of geographic distance. The gradients tested are expected to affect snake assemblages indirectly, such as clay content in the soil determining primary production and height above the nearest drainage determining prey availability, or directly, such as percentage of tree cover determining availability of resting and nesting sites, and climate (temperature and precipitation) causing physiological filtering. We sampled snakes in 21 sampling plots, each covering five $km^2$, distributed over 880 km in the central-southern Amazon Basin. We used dissimilarities between sampling sites to quantify TBD, FBD and PBD, which were response variables in multiple-linear-regression and redundancy analysis models. We show that patterns of snake community composition based on TBD, FBD and PBD are associated with environmental heterogeneity in the Amazon. Despite positive correlations between all β-diversity measures, TBD responded to different environmental gradients compared to FBD and PBD. Our findings suggest that multi-dimensional approaches are more informative for ecological studies and conservation actions compared to a single diversity measure.

## INTRODUCTION

Investigating how environmental gradients influence community structure is crucial to understanding mechanisms and processes affecting biodiversity at different scales

(*Keddy, 1992*). Quantifying species-habitat associations across continuous landscapes helps disentangle the mechanisms generating and maintaining patterns of regional and local biodiversity. This has been widely demonstrated in the Amazon rainforests by estimates of assemblage β-diversity associated with environmental gradients at mesoscales (*Drucker, Costa & Magnusson, 2008*; *Fraga, Lima & Magnusson, 2011*; *Bueno et al., 2012*; *Ribeiro, Lima & Magnusson, 2012*; *Rojas-Ahumada, Landeiro & Menin, 2012*; *Moulatlet et al., 2014*; *Menger et al., 2017*). In general, levels of β-diversity across heterogeneous continuous landscapes has been estimated through biotic complementarities and redundancies among sites. Most studies focused on measures of β-diversity based on between-site dissimilarities in quantitative (based on abundance data) and qualitative (presence/absence data) species composition. However, use of multiple dimensions may be more informative, because different diversity measures often carry complementary information (*Devictor et al., 2010*; *Weinstein et al., 2014*).

Numerous methods have been developed to quantify β-diversity, and each method potentially gives different insights into the mechanisms driving biodiversity (*Dehling et al., 2014*). Values of taxonomic β-diversity (TBD) may change across heterogeneous landscapes in response to variation in availability of resources, because of selection for different physiological characteristics, ecological plasticity, intra and interspecific interactions, and dispersal ability (*Mariac et al., 2011*; *Hangartner, Laurila & Räsänen, 2012*). In general, patterns of community assembly are caused by different portions of environmental gradients providing suitable conditions to habitat-specialist species, while generalists cover larger portions of gradients (*Kinupp & Magnusson, 2005*).

Estimates of functional β-diversity (FBD) may change across heterogeneous landscapes because function is mediated by phenotypes potentially affecting the species performance and fitness, such as morphological, biochemical, behavioral and phenological traits (*Petchey & Gaston, 2002*). Functional traits can be environmentally filtered when environmental heterogeneity is sufficient to cause variation in local adaptation to different selection pressures (*Weinstein et al., 2014*). Additionally, some traits such as foraging mode may determine the ability of snakes to cross different habitat patches in Amazonia (*Fraga et al., 2017*). High levels of FBD are often related to ecosystem dynamics, stability and productivity (*Tilman, 2001*).

Estimates of phylogenetic β-diversity (PBD) usually incorporate information on the evolutionary history that is shared among species throughout assemblages (*Milcu et al., 2013*). High values of PBD suggest convergent adaptation among historically isolated assemblages, but recently connected (*Weinstein et al., 2014*), which is expected as a result from timescale variation in stability and connectivity among habitat patches (*Morlon et al., 2011*; *Jetz & Fine, 2012*). In the Amazon rainforests, a combination between the Andean uplift and climate change are though as major factors driving habitat evolution, and consequently driving species diversification, dispersal and extinction (*Hoorn et al., 2010*). Alternatively, levels of PBD positively related to FBD suggest assemblages evolutionary structured by niche conservatism (*Wiens & Graham, 2005*).

Measures of β-diversity based on between-site differences in TBD, FBD and PBD are more effective at identifying factors shaping community structure than measures of
α-diversity, such as number of species or functional groups at particular sites (*Cadotte et al., 2009*; *Flynn et al., 2011*). This is because mechanisms influencing community assembly act on biotic complementarities and redundancies among sites, and not on the number of organism-units within sites (*Diaz & Cabido, 2001*; *McGill et al., 2006*). In general, α-diversity measures fail to capture the contribution of each species to the regional diversity, because different sites may have equal values of diversity (e.g., number of species), even if the species found in each site are taxonomically, functionally or phylogenetically distinct. Identifying mechanisms underpinning β-diversity has clear implications for conservation management. These include identification of unique characteristics in the regional diversity, which makes a site irreplaceable and therefore a priority for conservation actions (*Pressey et al., 1993*). This approach has been used to test the efficiency of protected areas in France (*Meynard et al., 2011*), the effects of forest modification on birds and trees in South Africa (*Grass et al., 2015*), and the efficiency of environmental law in Brazil (*Fraga, Lima & Magnusson, 2011*; *Bueno et al., 2012*).

Although much of the Amazon basin appears relatively homogeneous in satellite images, ecological studies at mesoscales have shown that subtle changes along environmental gradients influence patterns of co-occurrence of frogs (*Ribeiro, Lima & Magnusson, 2012*; *Rojas-Ahumada, Landeiro & Menin, 2012*), understory birds (*Bueno et al., 2012*; *Menger et al., 2017*), plants (*Costa, Magnusson & Luizão, 2005*; *Drucker, Costa & Magnusson, 2008*) and snakes (*Fraga, Lima & Magnusson, 2011*). In this study, we sampled a continuous landscape across about 880 km of rainforest, from central to southwest Amazonia. Most species of snakes recorded are widely distributed throughout the Amazon basin and some occur in other ecosystems in South America. However, species do not occupy all sites within their ranges, and different assemblages could be expected even at scales of tens of kilometers (*Fraga, Lima & Magnusson, 2011*). The wide distributions indicate that assemblage differences are more likely to result from environmental effects than from historical contingencies, such as dispersal limitation across geographic barriers.

In this study we examine the influence of environmental gradients on TBD, FBD and PBD estimates for snake assemblages in the Amazon rainforests. Investigating multiple assemblage dimensions in the same study system potentially allows accessing broad pictures of factors causing and maintaining biodiversity. Snakes are useful organisms to test for multidimensional changes in assemblages over landscapes in the Amazon because of the exceptionally-high species diversity (*Bernarde et al., 2012*), great species-trait diversity (e.g., body size and colors, reproductive and foraging modes), which implies large variation in functional traits (*Martins & Oliveira, 1999*), and heterogeneous habitats that potentially affect dispersal and gene flow (*Fraga et al., 2017*). In addition, snakes have been included in estimates of global reptile decline (*Gibbons et al., 2000*), which highlights the importance of assessing a broad picture of mechanisms generating diversity. This is particularly important in the area we sampled, because it is under strong anthropogenic pressure due to rapid urban growth (*Soares-Filho et al., 2006*; *Fraga et al., 2013a*), deforestation associated with paving the major access road
(*Fearnside & Graça, 2006*; *Soares-Filho et al., 2006*), and flooding by hydroelectric power plants (*Fearnside, 2014*). Few studies have attempted to identify multiple factors driving snake community assembly (*Fraga, Lima & Magnusson, 2011*). Furthermore, to our knowledge, the standardized sampling effort used over such a large area in this study is unprecedented.

We evaluate the effects of environmental gradients on snake assemblages and investigate spatial structuring causing levels of TBD, FBD and PBD among sites. We test the general hypothesis that patterns of community composition in taxonomic, functional and phylogenetic spaces result from current environmental heterogeneity. Additionally, we hypothesize that different β-diversity measures should respond to different environmental gradients, because they carry complementary information on snake assemblages.

## MATERIALS AND METHODS

### Snake sampling

We sampled snakes in 21 RAPELD sampling sites (*Magnusson et al., 2005*, *2013*), each of which has trails covering five km$^2$ (five km separated by one km). RAPELD is a Brazilian acronym to accommodate two study scales, rapid assessments and long-term ecological research (PELD—in Portuguese, *Pesquisas Ecológicas de Longa Duração*). RAPELD sampling sites will be mentioned throughout the text as modules.

In each module, we sampled 10 250 m long by 10 m wide plots with center lines following the altitudinal contours. The plots were distributed along two parallel five km long trails (five plots per trail) with standardized distance of one km between neighboring plots. The modules were distributed almost linearly over 880 km (Fig. 1) from central (Manaus, Amazonas) to southwest Amazonia (Porto Velho, Rondônia). The study area includes Central Amazonia, to the north of the Amazon River, the interfluve between the middle regions of the Madeira and Purus rivers, and the upper Madeira River, in southwestern Amazonia. Three modules were installed at the Ducke Reserve, which is a 100 km$^2$ fragment of non-flooded primary rainforest, located on the northern outskirts of Manaus. Eleven modules were installed along the federal highway BR-319 that connects Manaus to western Brazil. The highway was largely abandoned in the 1980s, and the modules were installed mainly to enable multi-taxa impact assessments of the effect of the road on biodiversity. Along this road, modules were placed in areas covered by primary and old-secondary rainforest, with patches of flooded forest and *campinarana* (forest on white sand). The southern Madeira River region contains seven modules. The Madeira River was recently dammed by two large hydroelectric power plants in the Porto Velho region, and the modules were installed along the banks of the river for monitoring the effects of flooding on biodiversity. The data used in this study were collected prior to flooding (see details below). The region is covered by primary and secondary rainforest under increasing threat of human occupation.

We sampled snakes by nocturnal active visual search, with two observers per plot, and standard searching time of 1 h per plot. In the Amazon, nocturnal search
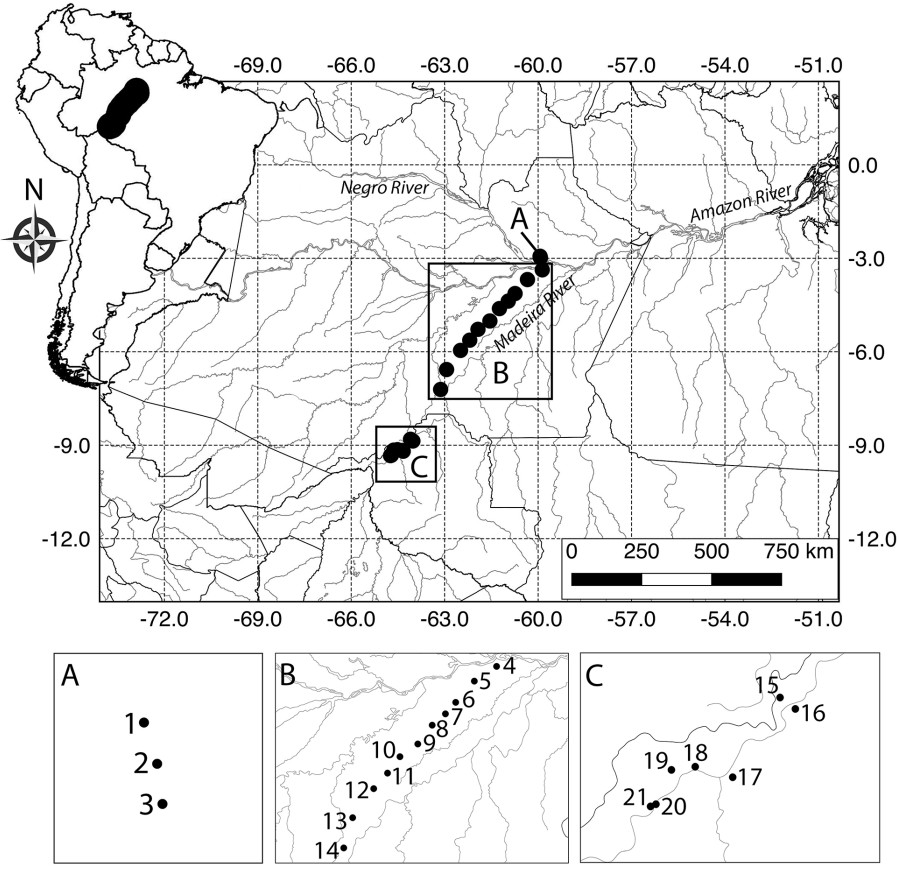

**Figure 1 Map of the study area in Brazilian Amazonia showing plots where snake assemblages were sampled.** Black circles are five km$^2$ RAPELD sampling modules (see definition of RAPELD in the methods). The numbers can be used to check the species found in each module in Table S1. (A) Ducke Reserve, Manaus. (B) Federal highway BR-319, Purus-Madeira interfluve. (C) Upper Madeira River, Rondônia.                                     

simultaneously allows finding foraging nocturnal species, and resting diurnal species (field observation). To increase the sampling effort and accuracy in species detection, we undertook four non-consecutive surveys of each plot between 2007 and 2014. We show below that temporal variation (e.g., caused by seasonal environmental variables) in assemblage structure is unlikely, because species composition did not differ among the surveys. Snakes were collected under RAN-ICMBio/IBAMA (Ministry of Environment, Government of Brazil) permanent license no 13777-2/2008 in the name of Albertina Pimentel Lima (coordinator of the sampling expeditions). The license includes ethical approval of all the procedures used.

Detection probabilities of snakes are usually very low (*Steen, 2010*; *Fraga et al., 2014*), and they may bias the results by generating statistical artifacts such as the arc effect (*Gauch, 1982*), mainly because no species occurrence is shared between sampling units. To avoid statistical artifacts in this study, we used modules as sampling units rather than plots. This results in loss of degrees of freedom, but it increases the predictive power of the analysis because the sampling units usually shared more than one species.

All the analyses in this study are based on 26 species (Table S1), belonging to four families (Boidae, Colubridae, Dipsadidae and Viperidae).

## Taxonomic β-diversity

We represented TBD as the scores from a Principal Coordinates Analysis (PCoA) ordination of a matrix of Forbes' similarity index (*Forbes, 1907*) on species presence/absence data between modules. We transformed Forbes' similarities in dissimilarities between modules by 1—Forbes. The Forbes index has been indicated as robust in the case of incomplete sampling (*Alroy, 2015*), which is common in studies of snakes in the Amazon due to the low detection probabilities of most species (*Fraga et al., 2014*). The PCoA was undertaken in the vegan package (*Oksanen et al., 2015*) in R (*R Development Core Team, 2015*).

## Functional β-diversity

We constructed a trait matrix using 10 continuous and discrete traits, measured or observed for adult individuals only. These were maximum total length, tail length proportional to body length, diameter of the eye proportional to head length, maximum size of offspring, discrete habitat (aquatic, arboreal, cryptozoic, fossorial, terrestrial), period of activity (diurnal, nocturnal), foraging mode (ambush, active), diet (birds, bird eggs, centipedes, earthworms, frogs, fish, lizards, mammals, snails, snakes, Squamata eggs, tadpoles), defensive behavior (ball posture, bite, caudal autotomy, cloacal discharge, constriction, enlarged head, flattened body, hidden head, inflated neck, liana imitation, shown mucosa, sound, strike, tail shaking, tail sting, venom, vomit) and reproductive mode (oviparous, viviparous). All the traits used have been described as ecologically relevant for snakes (literature compilation in *Burbrink & Myers (2015)*). Further details on functional traits may be found in Table S2.

The continuous traits were measured, and we used average values per species (see *Petchey & Gaston, 2006*). For the species for which we found less than five individuals, we supplemented our data with published data (*Beebe, 1946*; *Belluomini & Hoge, 1958*; *Duellman, 1978*; *Cunha & Nascimento, 1983*, *1993*; *Dixon & Soini, 1986*; *Michaud & Dixon, 1989*; *Starace, 1998*; *Martins & Oliveira, 1999*; *Fraga et al., 2013a*). We also obtained most of the data for discrete traits per species from the literature, and they were supplemented with field observations. The levels of most discrete traits are not mutually exclusive (e.g., species which feed on a variety of prey), so we coded discrete traits into independent binary traits as suggested by *Petchey & Gaston (2007)*.

We used the trait matrix to estimate FBD using the dbFD function in the functional diversity (FD) R-package (*Laliberté & Legendre, 2010*; *Laliberté, Legendre & Shipley, 2014*). This function calculates Gower distances between species, which is an index thought to be more appropriate when analyzing mixed continuous and discrete traits, although the results are often strongly correlated with Euclidean distances (*Petchey & Gaston, 2007*). The dbFD function transforms the Gower distance matrix by calculating square roots. This is important to avoid negative eigenvalues in the PCoA calculated from the distance matrix, which should be set in Euclidean space to avoid biased estimates of FD

(*Laliberté & Legendre, 2010*). PCoA was used to obtain scores representing four different functional indices (*Villéger, Mason & Mouillot, 2008*), which are functional richness, functional evenness, functional divergence and functional dispersion (FDis). In this study, we represented FBD using FDis, because this index estimates FBD based on average distances to the centroid of multivariate dispersion (*Anderson, 2006*). FDis has been described as a β-diversity index which is not affected by species richness, it can handle any number and type of traits and it is little biased by outliers (*Anderson, Ellingsen & McArdle, 2006*).

We visually controlled the robustness of the FBD estimate by constructing a functional tree (Fig. S1) based on Gower pairwise distances between species, which was calculated in the vegan R-package (*Oksanen et al., 2015*). We undertook a hierarchical cluster analysis on the Gower dissimilarity matrix to build an UPGMA functional tree, using the hclust function (argument average) in R.

## Phylogenetic β-diversity

We estimated PBD based on a well-supported phylogenetic hypothesis proposed by *Pyron, Burbink & Wiens (2013)*. Phylogeny of Squamata reptiles was reconstructed by analyzing 12 concatenated genes (five mtDNA and seven nuclear) from more than 4,000 species. Levels of clade support were estimated by non-parametric Shimodaira- Hasegawa-Like implementation of the approximate likelihood-ratio test (further details in *Pyron, Burbink & Wiens (2013)*). We used the APE R-package (*Paradis, Claude & Strimmer, 2004*) to obtain a subtree composed of the species sampled in this study (Fig. S2).

We estimated PD using the phylosor function of the picante R-package (*Kembel et al., 2010*), which estimates fractions of branch lengths in a phylogenetic tree that are shared among communities (*Kembel et al., 2010*). The phylosor function returns a pairwise phylogenetic similarity matrix, which was converted to a pairwise distance matrix (1-phylosor matrix) and summarized by PCoA scores.

## Environmental gradients and inferential analysis

We evaluated the influence of clay content in the soil on the diversity measures because this gradient affects primary production, which influences the overall trophic network (*Cintra et al., 2013*). Clay content was measured in a pooled five g sample derived from six subsamples per plot, and we used averages per module. Technicians at the Laboratório Temático de Solos e Plantas of the Instituto Nacional de Pesquisas da Amazônia (INPA, Manaus, Brazil) conducted the physical analyses following standard methods (*Empresa Brasileira de Pesquisa Agropecuária (Embrapa), 2009*).

The height above the nearest drainage (HAND) algorithm estimates the depth to the water table, which represents a gradient of relative water potential, in which higher values indicate large gravitational potential and lower values may reflect soil waterlogging in the absence of drainage. Here we tested the effects of HAND (log-normalized) on snake diversity because vertical and horizontal distances from drainage are correlated in a micro-watershed scale (*Schietti et al., 2013*), and horizontal distance from drainage was previously identified as an important factor affecting snake assemblages through
availability of physiological optimal or prey availability at assemblage (*Fraga, Lima & Magnusson, 2011*) and population (*Fraga et al., 2013b*) levels in Amazonia. We used percentage of tree cover (log-normalized) because this gradient potentially filters species by their adaptability to variation in habitat openness, considering the variation in factors such as availability of resting and nesting sites (*Burger & Zappalorti, 1986*), availability of prey and protection from predators (*Webb & Shine, 1997*), and light intensity (*Pringle, Webb & Shine, 2003*). The gradients HAND and tree cover were obtained from raster surfaces downloaded from the public repository Ambdata (*Amaral et al., 2013*; www.dpi.inpe.br/Ambdata).

We used temperature of the coldest month and precipitation of the wettest month because climate is often considered as a primary factor determining distribution of biodiversity at numerous spatial and temporal scales. Climate may limit species distribution through physiological filtering, especially in ectothermic animals such as snakes (*Blain et al., 2009*). Climate data were obtained from raster surfaces downloaded from the Worldclim database (*Fick & Hijmans, 2017*). All raster surfaces used in this study have a resolution of one km, and values for each gradient were extracted by the raster R-package (*Hijmans, 2015*) using geographic coordinates per module.

Geographic distance has been found to be an important factor driving β-diversity for several vertebrates, because potentially carries unmeasured environmental variation (*Qian & Ricklefs, 2012*). Our study area covers about 4,400 km$^2$, and therefore we expect spatial autocorrelation in the environmental data. To reduce the effects of geographic distance on the environmental gradients we calculated Euclidean distances between centroid coordinates per module, and reduced dimensionalities using the first axis from a PCoA. The coordinates were used in linear regressions given by *gradient = a + b (geographic distance)*. Residuals from these models were used as independent variables in inferential models to quantity proportions of snake diversity that are explained by environmental variation. We used this approach because multiple-linear-regressions using raw environmental data as independent variables returned slight spatial autocorrelation in the residuals for TBD (Moran´s $I = -0.272$, $P = 0.06$), FBD (Moran´s $I = -0.320$, $P = 0.02$) and PBD (Moran´s $I = -0.261$, $P = 0.08$), which was negatively significant ($P < 0.05$) in 20–30% of 10 geographic distance classes (Fig. S3).

To test for the effects of gradients on the diversity measures we used multiple-linear-regression models, that were built following the general formula *diversity measure = a + b$_1$(clay content residuals) + b$_2$(HAND residuals) + b$_3$(tree cover residuals) + b$_4$(temperature residuals) + b$_5$(precipitation residuals)*. The gradients measured are given in different units, so we scaled them using the scale function in R (*Becker, Chambers & Wilks, 1988*). The residuals representing gradients showed little multicollinearity ($\leq 0.7$ in all cases). We have considered significant relationships at $P \leq 0.05$ after Bonferroni correction. Additional information (e.g., amplitude and average values) on the gradients can be found in the Table S3.

Alternatively, we used redundancy analysis (RDA) to test the effects of environmental gradients on raw distance matrices among sampling units, separately for each diversity measure. This approach was useful to verify the robustness of the results obtained

**Table 1 Summary of statistical coefficients from multiple linear regressions testing the effects of environmental gradients on estimates of snake β-diversity in Brazilian Amazonia.**

|  | Coefficient | TBD | FBD | PBD |
|---|---|---|---|---|
|  | $R^2$ | 0.612 | 0.377 | 0.319 |
|  | $P$ | **0.001** | **0.02** | **0.05** |
| Clay content | SE | 0.007 | 0.01 | 0.002 |
|  | $t$ | −4.431 | −0.499 | 1.233 |
|  | $P$ | **0.002** | 1 | 1 |
| HAND | SE | 0.08 | 0.127 | 0.033 |
|  | $t$ | −0.045 | 3.325 | 3.142 |
|  | $P$ | 1 | **0.023** | **0.034** |
| Tree cover | SE | 0.16 | 0.247 | 0.064 |
|  | $t$ | 3.929 | −1.435 | 0.661 |
|  | $P$ | **0.007** | 0.859 | 1 |
| Temperature of the coldest month | SE | 0.23 | 0.035 | 0.009 |
|  | $t$ | −0.34 | −0.044 | −0.574 |
|  | $P$ | 1 | 1 | 1 |
| Precipitation of the wettest month | SE | 0.003 | 0.005 | 0.001 |
|  | $t$ | −1.804 | 1.382 | −0.589 |
|  | $P$ | 0.457 | 0.936 | 1 |

**Notes:**

Bolded $P$-values are statistically significant after Bonferroni correction. $R^2$ values are adjusted to the number of sampling units.

TBD, taxonomic β-diversity; FBD, functional β-diversity; PBD, phylogenetic β-diversity; HAND, height above the nearest drainage; SE, standard error.

by the multiple linear regressions, considering information loss by dimensionality compression through PCoA. We constructed the models using taxonomic, functional and phylogenetic distance matrices as dependent variables, and environmental gradients as independent variables, which is equivalent to multiple-multivariate linear regression models (*Legendre, Fortin & Borcard, 2015*). We used a permutation test of significance (5,000 randomizations) to decide between accept or reject null hypothesis.

We also compared the subsets of species that were found per plot in each of the four surveys (2007–2014), to control any effect of temporal variation on the diversity measures. We calculated Forbes pairwise dissimilarities among each survey on each plot, summarized the resulting matrix using PCoA (axis 1), and tested differences among surveys using ANOVA. We set a two-factors model—plot and survey—to decouple assemblage compositions between space and time. We found that differences in assemblage composition are associated with plots ($P = 0.03$), and plots and surveys interacting with each other ($P = 0.01$), but not with surveys alone ($P = 0.26$). This finding demonstrates that the patterns of spatial assemblage structure shown in this study have no bias of assemblages changing over time.

To quantify relationships between different diversity measures we used matrix regression with permutation test of significance (5,000 randomizations). The models were set up with the pairwise distance matrices used to summarize TBD, FBD and PBD.

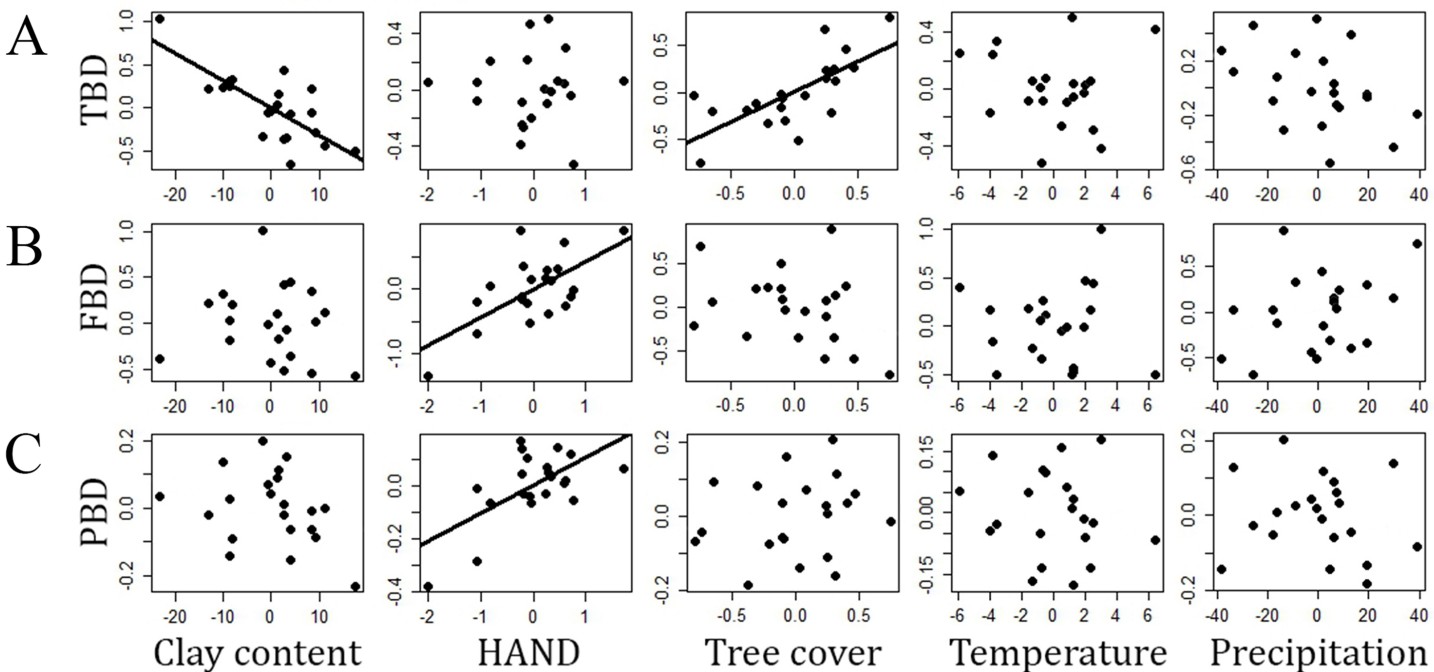

**Figure 2 Relationships (partials from multiple linear regressions) between different measures of snake β-diversity and environmental gradients in central-southwestern Brazilian Amazonia.** (A) TBD, taxonomic β-diversity, (B) FBD, functional β-diversity, (C) PBD, phylogenetic β-diversity. HAND, height above the nearest drainage. Environmental gradients are shown as residuals from linear regressions used to reduce the effects of geographic distance on the environmental heterogeneity measured.

## RESULTS

The first axis of PCoA ordination captured 59% of the original variance in the raw data used to estimate TBD, and the second axis captured 39%. However, the variance captured by axis 1 was not related to the environmental variation quantified by the gradients tested ($P > 0.25$ in all cases). Therefore, we used axis 2 to represent TBD. PCoA axis 1 captured 100% of the variance for FBD and 30% for PBD and were used as univariate versions of the diversity measures in the inferential models.

The multiple linear regression explained 61 percent (adjusted $R^2$) of the variance in TBD ($F_{5,15} = 7.31$, $P = 0.001$, residual standard error = 0.29). This finding was particularly associated to the effects of clay content ($P = 0.002$) and tree cover ($P = 0.034$) on β-diversity among the sampling modules. HAND, temperature and precipitation were not related to TBD ($P > 0.45$ in all cases). About 37% of the variance in FBD was explained by the multiple regression ($F_{5,15} = 3.42$, $P = 0.02$, residual standard error = 0.45), which was mainly due to the effects of HAND ($P = 0.023$) on dissimilarities between modules. Clay content, tree cover, temperature and precipitation were not related to FBD ($P > 0.85$ in all cases). About 32% of the variance in PBD was explained by the gradients ($F_{5,15} = 2.87$, $P = 0.05$, residual standard error = 0.11), but only HAND ($P = 0.034$) contributed to the model. The other gradients tested were not related to PBD ($P = 1$ in all cases). All the multiple regression models returned residuals that did not differ statistically from a normal distribution (Shapiro–Wilk $W > 0.92$, $P > 0.12$ in all cases). A complete summary

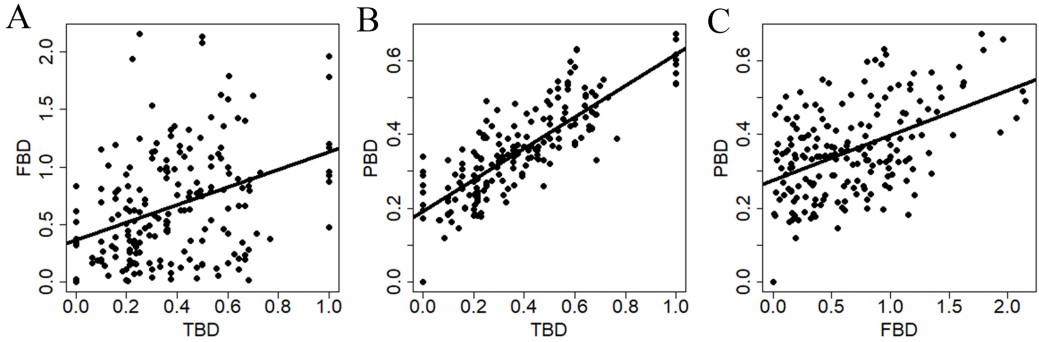

**Figure 3 Relationships between dissimilarity matrices used to summarize different estimates of snake β-diversity in central-southwestern Brazilian Amazonia.** TBD, taxonomic β-diversity; FBD, functional β-diversity; PBD, phylogenetic β-diversity. (A) Relationship between TBD and FDB. (B) Relationship between TBD and PBD. (C) Relationship between FBD and PBD.

of the multiple-regression models can be found in Table 1 and the partial regressions from each model are presented in Fig. 2.

The RDA models significantly captured 32% of the constrained variance to TBD ($F_{5-15} = 1.42$, $P = 0.05$), 43% to FBD ($F_{5-15} = 2.27$, $P = 0.04$) and 36% to PBD ($F_{5-15} = 1.71$, $P = 0.03$). In general, the results were consistent with the multiple regressions (Table S4). However, tree cover did not significantly explain TBD ($P = 0.24$).

All diversity measures were positively related to each other (Fig. 3). However, TBD was more strongly related to PBD ($r^2 = 0.67$, $P < 0.0001$) compare to the relationships between TBD and FBD ($r^2 = 0.13$, $P < 0.0001$), and between FBD and PBD ($r^2 = 0.23$, $P < 0.0001$).

# DISCUSSION

Our data indicate that environmental gradients affect snake co-occurrence in central-southern Amazonia, which results in levels of β-diversity identified by taxonomic, functional and phylogenetic dissimilarities among modules. These findings are consistent with temperate snake assemblages, which may be structured by phylogenetic species variability and trait variability (*Burbrink & Myers, 2015*). Furthermore, our data corroborate a previous study in Brazil, which found variation in phylogenetic and phenotypic compositions of snake assemblages associated with environmental gradients (*Cavalheri, Both & Martins, 2015*). However, the variation found in that study was primarily due to differences between forested and open biomes, which is expected to be pronounced. In this study, we show that variation in snake assemblages along environmental gradients is consistent even within biomes, where structural differences between sites are often subtle.

Environmental gradients affecting species composition have been found in many groups of organisms in the Amazon, such as frogs (*Ribeiro, Lima & Magnusson, 2012*; *Rojas-Ahumada, Landeiro & Menin, 2012*), understory birds (*Bueno et al., 2012*; *Menger et al., 2017*), plants (*Drucker, Costa & Magnusson, 2008*; *Moulatlet et al., 2014*) and snakes (*Fraga, Lima & Magnusson, 2011*). In general, it is expected that species occupy

portions of gradients in a way to optimize the balance between dispersal capacity, physiological needs and availability of resources (*Laliberté, Legendre & Shipley, 2014*). However, our data indicate that TBD is affected by different sets of environmental gradients compared to FBD or PBD, which suggests that patterns of spatial structure in snake assemblages in the Amazon may not be characterized by a single diversity measure. In fact, choosing environmental gradients as predictors in species-habitat association models is not a trivial task, although it is an effective approach to evaluate conservation issues such as environmental legislation (*Fraga, Lima & Magnusson, 2011*; *Bueno et al., 2012*).

We found that clay content in the soil predicted snake community assembly based on taxonomic dissimilarities among modules. Associations between soil texture and patterns of vertebrate community structure are often assumed as indirectly causal, because soil texture affects many factors driving regional species occurrence, such as vegetation density (*Woinarski, Fisher & Milne, 1999*) and distance from streams (*Bueno et al., 2012*). The environmental heterogeneity in the Amazon rainforests includes soil texture gradients from poorly drained, seasonally flooded sandy soils (*Cintra et al., 2013*), in which plants find poor substrate for rooting (*Quesada et al., 2012*), to well-drained soils that support older well-developed forests (*De-Castilho et al., 2006*; *Emilio et al., 2013*). Therefore, variation in soil texture across landscapes generates high β-diversity via suitability of the conditions for dispersal and colonization, which has been found in plant (*Costa, Magnusson & Luizão, 2005*; *Costa et al., 2008*), invertebrate (*Franklin, Magnusson & Luizão, 2005*) and vertebrate (*Woinarski, Fisher & Milne, 1999*; *Bueno et al., 2012*; this study) assemblages.

We showed that HAND is related to snake assemblages based on functional traits and phylogeny. Distance above the drainage has been identified as an important factor structuring plant (*Drucker, Costa & Magnusson, 2008*) and animal (*Fraga, Lima & Magnusson, 2011*; *Bueno et al., 2012*; *Rojas-Ahumada, Landeiro & Menin, 2012*) assemblages in the Amazon, and riparian zones may be biologically distinct from adjacent areas (*Sabo et al., 2005*). Despite the fact that most Amazonian snakes can cross different habitats, distance from drainage may influences β-diversity (*Fraga, Lima & Magnusson, 2011*) and structures populations in terms of variation in density and body size (*Fraga et al., 2013b*). In this study, we found that HAND affects FBD mainly because lower distances from drainage favor assemblages composed by arboreal (e.g., *Imantodes lentiferus*) and aquatic species (e.g., *Helicops angulatus*). Lower values of HAND are often associated with seasonal surface waterlogging (*Schietti et al., 2013*), which may complicate fossorial and terrestrial lifestyles. Therefore, variation in HAND across the landscape generates regional dispersal corridors that may be more suitable for some species than others, which could affect FBD and PBD.

We found inconsistency between the effects of tree cover on TBD estimated by multiple regression and RDA. Different results may reflect different levels of statistical sensitivity of the models to the data structure, or unidentified sampling bias. Therefore, we assume tree cover in this study as a probable filter to snake regional co-occurrence. Variation in tree cover along continuous landscapes generates mosaics of more and less

suitable habitats for different subsets of species, through factors that directly affects species biology. These include availability (*Lindenmayer et al., 1991*) and non-random selection of nesting and resting sites (*Burger & Zappalorti, 1986*; *Webb & Shine, 1997*), thermo-regulatory requirements, availability of prey, scent chemical trails from potential prey (e.g., small mammals), protection from predators (*Webb & Shine, 1997*), variation in light intensity, air and ground temperatures and wind speed (*Pringle, Webb & Shine, 2003*). Furthermore, variation in tree cover at wider spatial scales (e.g., biomes) may define regional subsets of species according to their morphological adaptation to use different plant strata (*Cavalheri, Both & Martins, 2015*). This finding is relevant for conservation, because maintenance of regional assemblages depends on protecting large areas, which contain wide amplitudes of tree-cover gradients.

Changes in phylogenetic (PBD) composition among plots were proportional to changes in the snake taxonomic identities (TBD). Spatial congruence between estimates of phylogenetic and TBD has been suggested as often higher compared to α-diversity measures (*Devictor et al., 2010*; *Bernard-Verdier et al., 2013*; *Arnan, Cerdá & Retana, 2017*), especially in cases of niche conservatism (*Wiens & Graham, 2005*). The tendency of species to retain ancestral characteristics along generations results in local or regional assemblages structured by environmental filtering, despite different environmental gradients may affect different diversity measures (*Webb et al., 2002*). Additionally, the positive correlations between diversity measures show that TBD and FBD efficiently captured a phylogenetic signal, which suggests assemblages evolutionary structured by Brownian motion, in which species change mainly through genetic drift and natural selection randomly directed (*Losos, 2008*). Concerning the positive relation between FBD and PBD, we found that environmental gradients drive co-occurrence of species that are simultaneously phylogenetically related and have similar ecological requirements (*Keddy, 1992*; *Myers & Harms, 2009*), and one measure may be used as a proxy to the other. Contrarily, TBD was not a good proxy for FBD, because large proportions of data were poorly fitted between both diversity measures. Ultimately, combinations of TBD and FBD should be part of studies on community ecology, because they give integrative approaches that reveal taxonomic, ecological and evolutionary forces acting on community structuring, which is very useful for conservation (*Devictor et al., 2010*).

From the point of view of conservation, the positive relationships between diversity measures directs the focus of biodiversity monitoring programs and reserve planning to cover higher levels of phylogenetic diversity, because this measure reflects the maintenance of ecosystem processes operating over long timescales (*Cadotte, Carscadden & Mirotchnick, 2011*). FD is estimated based on sets of traits that reflect environmental tolerances and requirements, which in turn determine where species can live (*Lavorel et al., 1997*) and interact with each other in assemblages (*Davies et al., 2007*). Therefore, loss of evolutionarily distinct species may result in irreversible loss of functions for ecosystems (*Bracken & Low, 2012*). However, at larger scales it may be difficult to decide which diversity measure should be prioritized in conservation, and a multi-dimensional approach may be more appropriate (*Devictor et al., 2010*), despite greater difficulties of interpretation.

Our findings are unlikely to be biased by the spatial distribution of sampling units (see *González-Caro et al., 2012*), because the RAPELD system provides regular distribution of plots across the landscape regardless of logistical issues (*Magnusson et al., 2013*). However, snakes usually have low detection probabilities (*Steen, 2010*; *Fraga et al., 2014*), which have been estimated at less than 10% for multiple surveys of many Amazonian species in RAPELD plots (*Fraga et al., 2014*). Low detection probabilities often cause false absences of species from plots, and this may generate misinterpretation of how species respond to landscape change (*Gu & Swihart, 2004*). We are unable to totally discount effects of low detectability on our results. However, the strong relationships between diversity measures and environmental gradients showed that a combination of high sampling effort, different methods used to quantify biodiversity and the use of an appropriate multivariate distance measure may considerably reduce the effects of false absences and return reliable results.

## CONCLUSIONS

We used an unprecedented standardized sampling effort to show that environmental heterogeneity is associated with β-diversity in Amazonian-forest snakes. Positive correlations between β-diversity measures estimated show that PBD alone may be sufficient to investigate spatial structure in Amazonian snake assemblages under taxonomic, functional and phylogenetic perspectives. However, TBD response to different environmental gradients suggests that testing the effects of a set of environmental gradients on at least two β-diversity measures can generate deeper understanding of factors causing spatial community assembly. This finding highlights the efficiency of using multi-dimensional approaches to quantify biodiversity in community-level conservation status assessments and decision-making on natural resources management.

## ACKNOWLEDGEMENTS

We thank E. Farias, M.C. Araújo, P.I. Simões, M. Antunes, D. Bower, Pinduca, Neneco, Rubico and Joãozinho for assistance in fieldwork. J.M.C. da Silva and two anonymous reviewers provided valuable comments on a previous version of the manuscript.

### Funding

Data collection in the Madeira River plots was supported by the Wildlife Conservation Program from Santo Energia S.A. Data collection in the Highway BR-319 plots was financed by the PRONEX—CNPq/FAPEAM, proc. 653/2009 to A.P. Lima. A major grant from the FAPESP/FAPEAM (465/2010) and CNPq (473308/2009-6) to HIDROVEG project and additional funding from provided by PRONEX—FAPEAM (1600/2006). PPBio—Programa de Pesquisas em Biodiversidade (CNPq 558318/2009-6), INCT CENBAM—Centro de Estudos Integrados da Biodiversidade Amazônica and LBA—Programa de Grande Escala da Biosfera-Atmosfera na Amazônia supported logistics to data collection. The Coordenação de Aperfeiçoamento de Pessoal de Nível

Superior—CAPES provided a scholarship to R. de Fraga. CNPq—Programa Ciência sem fronteiras provided scholarship to A. Stow. The funders had no role in study design, data collection and analysis, decision to publish, or preparation of the manuscript.

## Grant Disclosures

The following grant information was disclosed by the authors:
Wildlife Conservation Program from Santo Energia S.A.
Highway BR-319 plots was financed by the PRONEX—CNPq/FAPEAM, proc. 653/2009 to A.P. Lima.
FAPESP/FAPEAM (465/2010) and CNPq (473308/2009-6) to HIDROVEG project and additional funding from provided by PRONEX—FAPEAM (1600/2006).
PPBio—Programa de Pesquisas em Biodiversidade (CNPq 558318/2009-6).
INCT CENBAM—Centro de Estudos Integrados da Biodiversidade Amazônica.
LBA—Programa de Grande Escala da Biosfera-Atmosfera na Amazônia.
The Coordenação de Aperfeiçoamento de Pessoal de Nível Superior—CAPES provided a scholarship to R. de Fraga.
CNPq—Programa Ciência sem fronteiras provided scholarship to A. Stow.

## Competing Interests

The authors declare that they have no competing interests.

## Author Contributions

- Rafael de Fraga conceived and designed the experiments, performed the experiments, analyzed the data, contributed reagents/materials/analysis tools, prepared figures and/or tables, authored or reviewed drafts of the paper, approved the final draft.
- Miquéias Ferrão conceived and designed the experiments, performed the experiments, contributed reagents/materials/analysis tools, authored or reviewed drafts of the paper, approved the final draft.
- Adam J. Stow conceived and designed the experiments, contributed reagents/materials/analysis tools, authored or reviewed drafts of the paper, approved the final draft.
- William E. Magnusson conceived and designed the experiments, analyzed the data, contributed reagents/materials/analysis tools, authored or reviewed drafts of the paper, approved the final draft.
- Albertina P. Lima conceived and designed the experiments, contributed reagents/materials/analysis tools, authored or reviewed drafts of the paper, approved the final draft.

## Field Study Permissions

The following information was supplied relating to field study approvals (i.e., approving body and any reference numbers):

Specimens were collected under RAN-ICMBio/IBAMA permanent license number 13777-2/2008. ICMBio and RAN are institutes of the Ministry of Environment, Government of Brazil. These permits were subject to approval of all procedures for collecting and euthanizing snakes.

## Data Availability

All data can be freely downloaded at https://ppbiodata.inpa.gov.br/metacatui/#view/PPBioAmOc.240.3.

## Supplemental Information

Supplemental information for this article can be found online at http://dx.doi.org/10.7717/peerj.5628#supplemental-information.

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
