# Peer review of "Different environmental gradients affect different measures of snake β-diversity in the Amazon rainforests"

_PeerJ, doi:10.7717/peerj.5628_

## Round 0.1 · original submission · Major Revisions

Both reviewers suggested that your paper is a significant scientific contribution and fits well with PeerJ. However, they have also pointed out that the manuscript needs several improvements before publication.

They listed several points that you should address in a new version of the manuscript. Both reviewers indicated concerns about the methods that were used in your analyses and pointed out some ways to overcome them.

Please, when resubmitting your manuscript, carefully consider ALL points raised in the reviewers' comments, explain every change made, and provide proper rebuttals for any remarks not addressed.

Reviewer 1 ·

Basic reporting

The study is interesting and a good contribution to the field community structure at regional scales applied to snakes. The manuscript is well written and does not need to much revision in professional English. I made some suggestions in order to improve the quality of the manuscript, I have some criticisms related to some definitions in ecology, and also some criticism applied to the experimental design of this study. But it is something that could be easily tackled by the authors.

When I read the title of your manuscript I noticed you were studying the effect of “ecological gradients” on different measures of beta diversity. Nevertheless, I noticed that there are ambiguous uses to the term “gradient” in your manuscript. Sometimes you used the term gradient meaning “filter” (along the discussion), sometimes you used the term gradient meaning “scale” (e.g. line 275 – “the gradients geographic distance”, along the abstract,…). Gradient means a gradual variation of abiotic or biotic factors through space or time. By definition, a spatial scale cannot be considered a gradient like an environmental gradient is. The term “ecological gradients” used in the title might be being misused if you are considering space or time as gradients. Otherwise, the term “scale” or “filter” could be applied to variables other than the environmental. Scale could be applied to space, time, environment and a biotic factor. Filter could be applied in a similar way. As already there are a more efficient terminology that incorporates space and time and it is not the term gradient, I recommend some changes in the text in order to remove conceptual ambiguities applied to the term gradient.

Line 74 to 77. It seems to me that the mechanisms used to explain PD changes along landscapes are more associated to a population genetic/phylogeographic level (in other words, to an intraspecific level) than to a interspecific/community level. As the estimate of PD you used in your study is applicable to a community level, I recommend you to rewrite this part of the paragraph including explanations of changes in PD applied to a community level (e.g. extinctions, migrations, selection of lineages due to, habitat complexity, source and sink models in a metapopulation and metacommunity perspective,…) and remove explanations more applicable to a population genetics level (e.g. disruptive selection). I noticed more appropriate explanations for variations of PD at a community level in some paragraphs of your Discussion section and could be used in the introduction as well.

I tried to find the paper Magnusson et al. 2005 cited in lines 130-131, but I did not find It at the references. I’m afraid that some citations in the body of the manuscript are not at the references section. So I suggest revision of the references section in order to ensure that all of them are appropriately included.

I strongly recommend you to take the Fig. S1 off from the supplementary material and include it inside the body of the manuscript. This figure is really important for the comprehension of the study you conducted and for the RAPELD concept and experimental design. The inclusion of the figure inside the body of the paper would be more interesting for the comprehension of the readers.

HAND is a good acronym, but you included more than one sentence applied to it. Sometimes you used “vertical distance from drainage”, sometimes you used “height above the nearest drainage” (HAND). Please, use only one term and correct it along the text.

Experimental design

L. 156-157: “The sampling expeditions were distributed throughout the year so that samples are independent of seasonal variables, such as rainfall.” Have you statistically tested that the samples are independent to seasonal variables? It should be tested before the analyses you conducted.

Actually, this is an important issue to be considered in your analyses. As the title suggests, you tested different “ecological gradients” (meaning scales?) in order to test if there are changes in beta diversity estimates at a regional scales in the Amazon and I suppose you aimed to infer which variables/gradients where the most important for community structure . However, you only considered two ecological scales in your analyses: spatial and environmental. Why did you not include a temporal scale into your analyzes. If you did not include it because of what you mentioned in lines 156-157, you should conduct an analysis to test if there is a temporal effect in your data. As you made samplings from 2007 to 2014, I think a temporal scale should be somehow included into analysis.

About the explanatory variables you used into your multiple regressions: why you chose these variables and consider it good enough to explain TD, PD and FD differences among communities? For some of them I find good reasons (e.g. geographic distances, tree cover). For other ones, I think that other variables might be considered more important than those you used. For example, why you did not include climatic data as well (precipitation, average temperature, seasonality, or some bioclimatic variables from worldclim)? As the worldclim project provides several data for the Amazon and these data could be extracted in R through the package raster as you did with other variables, I think some bioclimatic variables could enrich your analyses. I have noticed that you do consider other variables (average temperature, total precipitation and seasonality) as important variables in your discussion (lines 330-333). So, why not to include them into your analyses?

Actually, I have one criticism to the inclusion of geographic distance as an explanatory variable into your multiple regressions, and not as a covariate. The title of your manuscript suggest that you were evaluating the influence of ecological gradients on beta diversity estimates applied to snakes. Therefore, the most important variables are those involving gradients. Space is not related to gradient, but it is applied to a scale. The same conclusion is applicable to time. I think you should reanalyze your data considering space and time as covariates, removing the effect of them from environmental variables. Doing so you could separate the effects of space, time and environment variations in beta diversity measures.

L. 163: You should include geographic distance as a covariate and reduce its effects in order to notice just the effect of environment variables structuring the beta diversity of snakes. You need to separate the spatial effect from the environmental effect of your analyzes.

L. 148 – savannah. It was mentioned that there are some savannah fragments nearby some RAPELD modules and it was also informed that they were not sampled. This is an important aspect to be considered, since it can increase habitat complexity and might influence TD, FD and PD. Could you include this information as an explanatory variable to your data? Have you already included information of habitat complexity and the presence of savannas in your modules with one of the variables you used in the study?
L. 216-230. Mitochondrial 12S sequences have fast evolution rates and provide too much variation to your data, even more at an interspecific level. Actually, it is interesting at some point. Nevertheless, at an interspecific level the evolution rates of 12S are so fast that uncertainties about mitochondrial evolution emerges. Evolutionary models are alternatives to tackle this problem, but there are more recommendations to improve the assessment of the PD. One of it is to include more than one marker (such as nuclear markers) into analysis, reducing artifacts by the evolution of “one simple gene” compared to the evolution of multiple markers in a consensus tree or species tree. Therefore, I recommend you to include more than one marker to calculate PD. How did you select the Tamura-Nei model of evolution as the best for your data? Have you run several models and selected it as the best? If so, please inform it at material and methods.
Another question to the same issue: why did you not use the branch length from a tree generated by other studies (e.g. Pyron et al. 2013. A phylogeny and revised classification of Squamata, including 4161 species of lizards and snakes. BMC Evolutionary Biology 13:93–146) to calculate PD? Piron et al (2013) tree already includes many markers an taxa, so the value of PD by this study seems better than those you generated only with 12S. At lines 231-233 you cited some studies (including the above mentioned) with apparently better phylogenetic hypotheses of snake evolutionary relationships just for comparison. Then, why did you not estimate the PD for each module by their studies instead of just citing them for comparison?
Results
You have conducted three separate multiple regression models, one for each diversity index (Taxonomic, Functional and Phylogenetic), but you did not include any correction to the α you used. Actually, you also did not specify which value of α you used for your analysis (I suppose that was 0.05). As you conducted three different analyses with the same basic data, I suggest you to include a correction index (such as Bonferroni or other one) to control your α previous to analysis and reduce type I errors. The same thing must be applied to the correlation analyses for the pairs of Beta diversity estimates (TDxFD; TDxPD; PDxFD) you conducted.

Validity of the findings

In conclusion, the paper presents a few things that need to be revised or redone to correct some conceptual and analytical misinterpretations. However, in general, the manuscript is well written and can be a good contribution to the study of community structure at the regional level, especially for snakes.

Additional comments

The study is interesting and a good contribution to the field community structure at regional scales applied to snakes. The manuscript is well written and does not need to much revision in professional English. I made some suggestions in order to improve the quality of the manuscript, I have some criticisms related to some definitions in ecology, and also some criticism applied to the experimental design of this study. But it is something that could be easily tackled by the authors.

Reviewer 2 ·

Basic reporting

This paper describes the beta-diversity of snake assemblages in Amazon rainforests using different dimensions of biodiversity (taxonomic, functional and phylogenetic). The introduction is clear (I suggested only minor changes) and well referenced. Because I am not a native speaker, I can not judge the English quality. The figures and legends are clear and also need only minor changes. The authors provided the raw data which allowed me conduct some tests that I specify below.

Experimental design

The study is well conducted and the sampling effort is impressive. The authors performed a standardized sampling of snakes in a large area of the Amazon. I really appreciate that they care and consider the limitations of snake detectability in their analyses. The authors provided two field study permits but I think they could provide a bit more information about these documents to check their appropriateness. The research question is well defined. I only suggested to specify a little bit the hypothesis to not be so general. I also suggested to the authors some additional tests to confirm their results, especially applying a multiple regression on distance matrices to confirm the results of the summarized multiple linear regression. I got some results slightly different from those reported in the text when reproducing the analyses, and I suggested to the authors to check if the raw data available is correct (if it is exactly the same used in the paper) or if there is any method description missing (see details below).

Validity of the findings

In analyzing the data, I observed collinearity among some predictor variables. Such collinearity may be inflating the explained variance of the taxonomic beta-diversity and altering its result. The result that I got after removing one collinear variable is similar to that observed after applying a multiple regression on distance matrices. I recommend to the authors check if the raw data available is correct to confirm such effect. If the collinearity is real, great part of the discussion as well as the conclusion of the paper will need to be altered (see details below).

Additional comments

Major comments:
- Lines 126-127: The hypothesis is interesting but it seems too general. Consider thinking in a more objective hypothesis, something like evaluate if all the three measures of beta-diversity respond to the same environmental gradient or, alternatively, each measure has its own environmental predictor.
- The first field study permit provided is actually an operating license for an energy company. The document does not seem describe permission about snake sampling. I am not familiar with this kind of document. Can you clarify this permit? Additionally, its date of emission seems posterior to the dates of collection (2016). Is that right? The number of second field study permit I think is incomplete. Would it be 13777-2/2008? It is necessary an authenticity code and date of emission to check the appropriateness of the document. Can you, please, provide some additional information?
- You summarize each measure of beta-diversities by applying an ordination method and using one of the resulting scores as response variable in a multiple linear regression. But you did not inform how much each of the used scores is summarizing the respective measure of beta-diversity. For taxonomic beta-diversity, for example, it seems that you used the second score instead of the first (TD2 in the raw data). Is TD2 more informative then TD1? I think that would be interesting provide a table in the supplementary material informing for each measure of beta-diversity how much of its variability is summarized in each (two or three) score and indicate which score was used in the analyses. In addition, using scores necessarily implies loss of information. Especially depending on how much information is summarized by the used scores. Therefore, I think that would be interesting repeat the tests using a multiple regression on distance matrices just to confirm the results.
- In the lines 231-234 you report that there was low support for several branches in the phylogeny, but that the overall phylogeny did not conflict significantly with other existing and well-supported phylogenetic hypotheses. Do you think that would be interesting repeat your tests using these well-supported phylogenetic hypotheses available just to confirm the robustness of your results?
- In the line 267 it is affirmed that the gradients showed little collinearity. However, using the raw data available I found strong correlation between geographic distance and clay content (r = 0.7). The collinearity was confirmed using the VIF analysis and the Klein’s rule (see the function imcdiag in the mctest R package). But I may have overlooked any method to extract the scores from geographic distance (I calculated the Euclidian distance based on the coordinates available in the raw data and used the first axis of the PCoA) or the raw data available may be slightly different from that used in the paper. Please, check the collinearity in your data to confirm that they are not affecting the results. If the collinearity is real it will have great impact in your results (see below).
- I reproduced the analyses and found collinearity between geographic distance and clay content. When I removed one of these variables the result for TBD changed. The results showed that no environmental variable explain taxonomic beta-diversity (score TD2). The closest variable was HAND, with p = 0.0697 after removing clay content. Curiously, applying the multiple regression on distance matrices (which not reduce the information to one score) returned the same result. This may indicate that HAND is the best (only) predictor variable for all three beta-diversity measures in the area analyzed.
- Lines 319-321: You state that different measures of beta-diversity such as FBD and PBD are affected by different ecological gradients. However, these two measures were equally affected by the same environmental variable: HAND. Do you think that the additional effect of geographic distance on PBD (which must be revised; see below) is enough to affirm that both measures are affected by different sets of ecological gradients?

Minor Comments:
- Line 31: “Snake assemblage turnover”? You did not mention the taxonomic group in the hypothesis.
- Line 31-33: It is not clear in the abstract why variables such as clay content or HAND could influence beta-diversity of snakes. Consider provide some additional information first, something like: “…are influenced by gradients of geographic distance, as well as by environmental variables associated with… (the reason why you chose these three variables)”.
- I think that would be more interesting abbreviate the beta-diversity measures as TBD, FBD and PBD instead of TD, FD and PD to avoid confusion with measures of alpha diversity.
- I think you should avoid using turnover to report your results and use just beta-diversity (e.g., lines 31, 303). As you surely know, beta-diversity is composed by two distinct components (turnover and nestedness), and you did not partition the overall beta-diversity to state about the turnover component.
- line 62: Please, check if it is missing a comma after “resources”.
- It is not essential but consider moving forward the paragraph starting at line 80. That is, experiment change the position of the fourth and fifth paragraph to make your introduction more linear.
- Consider moving the lines 134-137 to the next paragraph (line 154). This way you will have one paragraph describing the modules and another paragraph detailing the sampling.
- Line 152: include the expression “(see details below)” after “prior to flooding”. Just to notify the reader that information about sampling date will be detailed soon.
- Just for curiosity. Do you think that the period of sampling (nocturnal; line 154) may have any influence in the period of activity trait (diurnal, nocturnal; line 180)?
- Lines 231-234 discusses about the phylogenetic tree. I think they should be moved to the line 226, before describing the phylogenetic beta-diversity.
- Line 237: Are these coordinates the centroid of the modules? Please, clarify.
- Coordinates information has some differences between files provided. Please check and standardize the data.
- HAND and Tree cover were log transformed but this is not mentioned in the text.
- Line 279-others: You provide the R-squared of the regression. But I think you should provide the adjusted R-squared since you are calculating a multiple regression.
- Please, check the p-value numbers. In some cases the number does not seem rounded correctly (e.g., the p-value for effect of Tree cover on PBD is 0.958 and not 0.957).
- Line 288: Geographical distance contributed to explain PBD (p = 0.05). However, my analyses returned p = 0.066. Please, check if the raw data available is correct.
- Line 297: The relationship between TBD and the two others beta-diversity measures informed in the text are different from those calculated with the data available. For example, the test for relationship between TBD and FBD returned r = -0.39, p = 0.08, and not r = 0.1, p = 0.1 as informed. Please, check if the data available and your results are correct. You could also think about applying a mantel test using the dissimilarity matrices.
- In the figures legend sometimes is wrote beta in others used the Greek letter. Please, keep a pattern.
- I could not reproduce exactly the graphics involving PBD. But just some points (e.g., the upper left points of the TBD x PBD plot), nothing affecting the relationship informed. In addition, the axis of the PBD in the figure 2 (ranging from -0.4 to 0.2) are different from the data provided (0.5 to 1.3). Please, check if the PBD score provided is the same used in your analyses.
- Line 302: “Our results indicate that…”
- Line 429: “multidimensional”?

---

## Round 0.2 · Minor Revisions

I have received the comments from two reviewers. I am glad to report that they are fine with the new version of your paper. However, they have still a few concerns that you should address before your article is ready for publication. Please, work on ALL these concerns and send your updated version as soon as you can.

Reviewer 1 ·

Basic reporting

I analyzed the response letter from Fraga et al. and I also read the new version of the manuscript. In general, the authors attempted to solve the problems presented in the first version of the manuscript, which made them rethink the introduction and analyses. They made a good job reanalyzing data and incorporating the suggestions from the first draft. Nevertheless, I still have some small suggestions to the authors, prioritizing cohesion and fast reading.

It seems that there are some words missing along the text. Please, make a revision mainly at lines 33, 88, 135 (“spaces”).

I noticed that you used the term ecological gradient (abstract) and environmental gradient (along the text) with the same meaning. Please, choose one of them to uniformize the text.

The second paragraph seems dislocated in the introduction, since you already expose your objectives, and not the general question to be studied. I think it would be more appropriated if it was placed at the last paragraph of the introduction. Similarly, part of the last paragraph of the introduction might be more useful if it was dislocated to the second paragraph.

One of your conclusions is not associated with your results. At lines 455-458 you conclude: “Our findings highlight the importance of investigating different diversity measures and ecological gradients in ecological studies and conservation actions, because multidimensional approaches are clearly more informative than simple metrics, such as the number of species that occur in a site.” First, I do agree with this sentence in a broad vision. However, your results (lines 337-339) indicate that all diversity measures were positively related to each other. Moreover, it was indicated at the results that the three measures of diversity used in the study were associated with HAND or clay content and three cover. These results make me conclude that, due to the strong association among diversity measures, the inclusion of just one or two of them would bring similar results you got at your study. Moreover, of the variables you selected, just three of them were considered significantly important. Therefore, your results are not related to this conclusion. Actually, I suggest you to make a discussion paragraph about it, taking it off the conclusion.


Table 1. Change precipitation and temperature to the variables you properly used (temperature of the coldest month and precipitation of the wettest month).

Experimental design

The issues emerged about the material and methods in the first draft were properly tackled by the authors. I do not have any criticism applied to this new version.

Validity of the findings

In general, I do not have anything important to comment about the results of the study. Just that one of your conclusions is not associated with the results of the study. I made a comment about it in the basic reporting section.

Additional comments

I analyzed the response letter from Fraga et al. and I also read the new version of the manuscript. In general, the authors attempted to solve the problems presented in the first version of the manuscript, which made them rethink the introduction and analyses. They made a good job reanalyzing data and incorporating the suggestions from the first draft. Nevertheless, I still have some small suggestions to the authors, prioritizing cohesion and fast reading. These suggestion can be easily solved by the authors, reason why I recommend minor revisions to the manuscript.

Reviewer 2 ·

Basic reporting

The revisions adequately addressed my concerns.

Experimental design

The revisions adequately addressed my concerns.

Validity of the findings

The revisions adequately addressed my concerns.

Additional comments

A few minor comments below.

- Line 359-361 may give the impression that diversity measures strongly correlated are affected by the same environmental gradients, which is not the case (e.g., TBD and PBD). Consider replace “weakly correlated” by “different”.

- The acronyms used in the Table 1 are different from those used in the text (BTD x TBD). Standardize it.

- The legend of the Fig. 3 must be revised. For example, the plot B, which show the relationship between PBD and TBD, is labeled as “FBD = Functional B-diversity”.

- Just an observation. I understand that you used the residuals of the environmental variables to remove the effects of geographical distance. In this case you are analyzing only the pure environmental factor on community beta-diversity, and it does not seem wrong since it is explicit in the text. However, keep in mind that environmental variables are spatially structured and by using this technique you are reducing their power of explanation. What I mean is that environmental predictors will have spatial autocorrelation, but that is not a problem. The problem only occurs if there is spatial autocorrelation in the residuals of your model. I believe that a more appropriate solution would be run the model with your environmental predictors and check the residuals. If there is no spatial autocorrelation in the residuals (which is your case, I checked) you don’t have to worry about geographical effects on your results.

---

## Round 0.3 · accepted · Accept

I have reviewed all the changes that you have made in the last version of your article and I think your paper is ready for publication. Congratulations again!

#